# Interaction of Cerium Oxide Nanoparticles and Ionic Cerium with Duckweed (*Lemna minor* L.): Uptake, Distribution, and Phytotoxicity

**DOI:** 10.3390/nano13182523

**Published:** 2023-09-08

**Authors:** Yang Liu, Xuepeng Zhao, Yuhui Ma, Wanqin Dai, Zhuda Song, Yun Wang, Jiaqi Shen, Xiao He, Fang Yang, Zhiyong Zhang

**Affiliations:** 1Hebei Provincial Key Laboratory of Green Chemical Technology & High Efficient Energy Saving, School of Chemical Engineering and Technology, Hebei University of Technology, Tianjin 300130, China; liuy99@ihep.ac.cn (Y.L.); zhaoxuepeng@ihep.ac.cn (X.Z.); 2Key Laboratory for Biomedical Effects of Nanomaterials and Nanosafety, Institute of High Energy Physics, Chinese Academy of Sciences, Beijing 100049, China; daiwanqin@ihep.ac.cn (W.D.); songzd@ihep.ac.cn (Z.S.); yunwang@ihep.ac.cn (Y.W.); shenjq@ihep.ac.cn (J.S.); hexiao@ihep.ac.cn (X.H.); 3School of Nuclear Science and Technology, University of the Chinese Academy of Sciences, Beijing 100049, China

**Keywords:** CeO_2_ nanoparticles, duckweed, uptake, distribution, toxicity, transformation

## Abstract

As one of the most widely used nanomaterials, CeO_2_ nanoparticles (NPs) might be released into the aquatic environment. In this paper, the interaction of CeO_2_ NPs and Ce^3+^ ions (0~10 mg/L) with duckweed (*Lemna minor* L.) was investigated. CeO_2_ NPs significantly inhibited the root elongation of duckweed at concentrations higher than 0.1 mg/L, while the inhibition threshold of Ce^3+^ ions was 0.02 mg/L. At high doses, both reduced photosynthetic pigment contents led to cell death and induced stomatal deformation, but the toxicity of Ce^3+^ ions was greater than that of CeO_2_ NPs at the same concentration. According to the in situ distribution of Ce in plant tissues by *μ*-XRF, the intensity of Ce signal was in the order of root > old frond > new frond, suggesting that roots play a major role in the uptake of Ce. The result of XANES showed that 27.6% of Ce(IV) was reduced to Ce(III) in duckweed treated with CeO_2_ NPs. We speculated that the toxicity of CeO_2_ NPs to duckweed was mainly due to its high sensitivity to the released Ce^3+^ ions. To our knowledge, this is the first study on the toxicity of CeO_2_ NPs to an aquatic higher plant.

## 1. Introduction

Due to their unique physicochemical properties, the applications of nanoparticles (NPs) are growing rapidly. Among various NPs, cerium oxide (CeO_2_) NPs have been widely used in a variety of industry applications, such as in catalysts, polishing agents, fuel additives, etc. [1,2,3]. The global production of CeO_2_ NPs was estimated to be about 10,000 tons per year [4]. Along with manufacture, transportation, use, and disposal, NPs (including CeO_2_ NPs) are inevitably released and accumulated in the environment. Therefore, there is growing interest in studying the fate of NPs in the environment and assessing their impact on the environment and biota [5,6,7,8].

The aquatic environment is an important sink of NPs, including CeO_2_ NPs. The predicted environmental concentrations (PEC) of CeO_2_ NPs in fresh water and sediments were 0.51–25.59 ng/L and 0.34–253.49 μg/kg in 2017, respectively [9]. Moreover, the maximum concentration of CeO_2_ NPs in sewage treatment effluent was predicted to be 1.87 mg/L [9]. Aquatic plants are an important part of the aquatic ecosystem. They are not only the habitat and food for aquatic animals but also a source of nutrients. Currently, there are few studies on the effects of CeO_2_ NPs on aquatic plants, mainly focusing on unicellular algae. The EC_50_ (50% effective concentration) values of CeO_2_ NPs to the freshwater microalgae ranged from 4.1 to 29.6 mg/L according to several studies, and the inhibitory effects were mainly caused by membrane damage due to direct contact or indirectly induced reactive oxygen species (ROS) generation [10,11]. 

Duckweed (e.g., *Lemna minor* L.) is a small floating aquatic plant which has the physiological characteristics such as small size, high reproduction rate, asexual reproduction and sensitivity to certain chemicals, etc. It has been widely used as a model for environmental monitoring and toxicity testing [12,13,14,15]. There are currently several studies on the interaction between some NPs and duckweed [16,17,18,19]. For instance, Ag NPs (6.84 mg/L) could change cellular ultrastructure, reduce photosynthetic pigments and starch grains along with the Ag accumulation in duckweed (*Landoltia punctata*) [16]. After exposure to Ag/Ag_2_S NPs and AgNO_3_, Ag_2_S NPs remained as Ag_2_S in duckweed (*Landoltia punctata*), while Ag NPs was transformed to Ag_2_S and silver thiol species and AgNO_3_ was transformed to Ag and sulfur-associated Ag species in tissues [17]. The growth of the duckweed species *Landoltia punctata* was not strongly affected by 1000 mg/L CuO NPs, but the reduction in photosynthetic pigments and the destruction of cellular structure were observed [18]. In another study, 150 μg/L CuO NPs altered the frond histomorphology and root cell integrity of duckweed (*Lemna minor*). Moreover, the authors confirmed that root uptake was the main pathway for the internalization of CuO NPs and chloroplasts were the sites of ROS production [19]. Unfortunately, there is no report on the biological effects of CeO_2_ NPs on duckweed. Given the possible negative impacts of CeO_2_ NPs on environmental species, it is very important to address this information gap in aquatic ecotoxicology. 

In the current study, we aimed to study the uptake, distribution, and toxicity of CeO_2_ NPs and Ce^3+^ ions to duckweed (*Lemna minor* L.) and to discuss the possible mechanisms. A range of endpoints such as plant growth, photosynthetic pigment contents, and cell death were assessed. The distribution of Ce in duckweed tissues was analyzed using micro-X-ray fluorescence (*μ*-XRF). The transformation of CeO_2_ NPs was investigated using X-ray absorption near-edge spectroscopy (XANES), and the contribution of NP-dissolution to the toxicity of CeO_2_ NPs to duckweed was estimated. Our results will provide further insight on the mechanisms underlying the impacts of metal-based NPs on aquatic species.

## 2. Materials and Methods

### 2.1. Material Preparation and Characterization

CeO_2_ NPs were synthesized via a precipitation method as previously described [20]. All chemicals were analytical grade and obtained from Aladdin Reagent Co., Ltd. (Shanghai, China). The synthesized CeO_2_ NPs were observed using scanning electron microscopy (SEM, Hitachi S-4800, Tokyo, Japan) and transmission electron microscopy (TEM, Tecnai G2 20 S-Twin, FEI, Tokyo, Japan). The hydrodynamic diameter and zeta potential of CeO_2_ NPs were determined using dynamic light scattering (DLS, ZetaSizer, Malvern Instruments, Malvern, UK) analysis. 

### 2.2. Plant Cultivation and Exposure

Duckweed (*Lemna minor* L.) was collected from a wetland (40 N latitude, 116 E longitude) in the southwest of Beijing, China and then washed with deionized water and submerged in 0.025% sodium hypochlorite solution for 1 to 3 min to remove algae and other impurities. The plants were pre-cultured in 1/10 modified Steinberg’s medium for one month under laboratory conditions according to the Organization for Economic Co-operation and Development (OECD) 2006 protocol [21] with a 16 h photoperiod in fluorescent light (8000 lux) and 8 h in the dark at 24 ± 2 °C. For exposure, the two-fronded duckweed with similar size were selected and incubated in CeO_2_ NP suspensions containing 1/10 modified Steinberg nutrition medium. In each treatment, 15 duckweed plants with their roots cut off were placed on a Petri dish. The duckweed was exposed to 0, 0.02, 0.1, 0.5, 2, and 10 mg/L of CeO_2_ NPs and their ionic counterpart Ce^3+^ ions (obtained by dissolving Ce(NO_3_)_3_ in water), respectively. Each treatment was replicated at least three times. All analyses were performed after 3 days of exposure.

### 2.3. Growth Related Parameters

At the end of experiment, the plants were thoroughly washed with deionized water and blotted with paper towels. The root lengths and fresh weight (Fw) of the plants were measured for each Petri dish. After weighing the Fw, the duckweed was oven-dried at 60 °C to a constant weight, and the dry weight (Dw) was determined. Photosynthetic pigment (chlorophyll a/b, total chlorophyll, and total carotenoid) contents were measured, and the detailed methods were described in Appendix A. The average specific growth rate (r) was calculated according to the method shown in the Appendix A.

### 2.4. Cell Death and SEM/TEM Observation

After treatment with 10 mg/L CeO_2_ NPs and Ce^3+^ ions for 3 days, the cell death in the frond and root of duckweed under different treatments was visualized using Evans blue (Eb) staining according to the previous methods with some modification [22,23,24]. The detailed methods were described in Appendix A. 

After treatment with 10 mg/L CeO_2_ NPs and Ce^3+^ ions for 3 days, the morphology changes of duckweed were observed using SEM (S4800, Hitachi, Tokyo, Japan), and the distribution of Ce-containing materials in tissues was observed using TEM (JEM-1230, JEOL, Tokyo, Japan). The samples were prepared as described in our previous report [11]. 

### 2.5. Ce and Mineral Elements Analysis

The bioaccumulation of Ce and mineral elements by duckweed under different treatments was determined. The lyophilized dried samples were ground into powders and digested with a 4:1 (*v*/*v*) mixture of concentrated HNO_3_ and H_2_O_2_. The elemental content of Ce was analyzed by inductively coupled plasma mass spectrometry (ICP-MS, Thermo Elemental X7, Waltham, MA, USA). Mineral elements (Ca, Mg, Fe, Mn, Zn) were analyzed using ICP-optical emission spectrometry (OES, Perkin Elmer Optima 8000, Waltham, MA, USA). 

### 2.6. Elemental Mapping by μ-XRF

Duckweed treated with 10 mg/L CeO_2_ NPs and Ce^3+^ ions were washed thoroughly with deionized water and were lyophilized. The dried samples were fixed on 3M tape and the *μ*-XRF experiments were performed at 4W1B beamline of Beijing Synchrotron Radiation Facility (BSRF, Beijing, China). The energy of the storage ring was 2.5 GeV, and the current intensity was 150 to 250 mA. The spot was focused to 20 × 50 μm by the polycapillary lens. The sample was held on a precision motor-driven stage, and the elemental 2D mapping was acquired in a stepwise manner with 50/100 μm steps. The data were analyzed using the PyMca package, and the elemental mappings were created using Origin.

### 2.7. Speciation of Ce in Duckweed

To explore the speciation of Ce in duckweed treated with 10 mg/L CeO_2_ NPs and Ce^3+^ ions, XANES analysis was performed, and samples were prepared as previously described [25]. Ce L_III_-edge spectra were collected at the 1W1B beamline of BSRF (Beijing, China). CeO_2_ and CePO_4_ were used as standard compounds, and their spectra were collected using transmission mode. The spectra of samples were collected using fluorescence mode. XANES spectra were normalized using Athena software and the speciation of Ce in samples was analyzed by linear combination fitting (LCF).

### 2.8. Statistical Analysis

Data were given as the mean ± standard deviation (SD). All statistical analyses were performed on the Statistical Package for Social Sciences (SPSS) version 22.0. One-way analysis of variance (ANOVA) and Tukey’s HSD post hoc test were used to check for statistically significant differences at *p* < 0.05.

## 3. Results

### 3.1. Characterization of CeO_2_ NPs

The TEM and SEM images of CeO_2_ NPs are shown in Appendix A. The average size of the NPs was calculated to be 7.3 ± 1.1 nm. XRD spectrum showed that CeO_2_ NPs have a cubic fluorite structure. The hydrodynamic particle sizes of CeO_2_ NPs (10 mg/L) in deionized water and Steinberg’s medium were 71.7 ± 0.5 nm and 306.1 ± 15.6 nm (Appendix A), respectively, indicating that CeO_2_ NPs was easier to agglomerate in the culture medium. After exposure, the hydrodynamic particle size of CeO_2_ NPs increased to 723.8 ± 106.1 nm (Appendix A). The zeta potential of CeO_2_ NPs suspensions changed from 32.4 ± 1.5 mV in deionized water to −14.7 ± 1.8 mV in Steinberg’s medium. 

### 3.2. Effect of CeO_2_ NPs and Ce^3+^ Ions on the Growth of Duckweed

After exposure to CeO_2_ NPs or Ce^3+^ ions, the root length, frond area, and biomass of duckweed are shown in Figure 1A. Compared with the control, the root elongation was significantly inhibited under the two treatments (except 0.02 mg/L CeO_2_ NPs). Moreover, the root length of Ce^3+^ ions treatment was significantly shorter than that of CeO_2_ NPs treatment at each exposure concentration. At 10 mg/L, the inhibition rate of Ce^3+^ ions to duckweed was 96%, and the root hardly developed. The average specific growth rate was calculated using the frond area of duckweed (Figure 1B). Similar to the result of root elongation, the average specific growth rate of duckweed fronds decreased with increasing concentrations of exposure materials. At the highest concentration, the average specific growth rates of duckweed in the NP group and the ionic group were 21.0% and 6.3%, respectively. The EC_20_ values of CeO_2_ NPs and Ce^3+^ ions for duckweed were calculated to be 1.78 mg/L and 0.39 mg/L, respectively, indicating that the latter was more toxic than the former.

Biomass is an important parameter related to photosynthetic efficiency. Fw and Dw as functions of concentration are shown in Figure 1C,D. Similarly to the results of root length and frond area, Fw and Dw of duckweed treated with CeO_2_ NPs and Ce^3+^ ions were not affected at lower concentrations but significantly decreased at concentrations higher than 0.1 mg/L. By comparing the apparent toxicity indexes, it was found that the effect of the highest concentration of 10 mg/L CeO_2_ NPs on duckweed was approximately comparable to that of Ce^3+^ ions at 0.5–2 mg/L. 

### 3.3. Pigments and Mineral Elements Analyses

The photosynthetic damage caused by CeO_2_ NPs and Ce^3+^ ions was analyzed based on the contents of photosynthetic pigments (Appendix A). At concentrations higher than 0.5 mg/L, the contents of chlorophyll a, chlorophyll b, and carotenoids in the fronds of duckweed under the two treatments were significantly reduced compared to the control, which was consistent with the result of biomass. At the end of the experiment, the fronds of duckweed treated with CeO_2_ NPs (10 mg/L) showed chlorosis, while the fronds of the ion-treated group grew more slowly and showed severe chlorosis, and the edges of the fronds became transparent (Appendix A). 

The contents of mineral elements, including macro elements (Ca and Mg) and trace elements (Fe, Mn, and Zn), in duckweed under different treatments are shown in Table 1. With the increase in exposure concentration, Ca contents in duckweed decreased significantly in both treatments, while Mg contents were almost unchanged (except 10 mg/L Ce^3+^ ions). In the NP group. Fe contents in duckweed gradually decreased with the increase in concentration. Different from this, Fe contents in the ionic group were significantly reduced compared to the control, but there was no concentration-dependent effect. Mn contents in duckweed exposed to different concentrations of CeO_2_ NPs showed a similar decreasing trend to the Fe contents, while Mn contents in the ionic group did not change significantly (except 10 mg/L Ce^3+^ ions). The change in Zn contents in duckweed as similar to that of Fe. 

### 3.4. Visualization and Quantification of Plant Cell Activity

The cell membranes of plants are selective, and the dye can be excreted by living cells when plant tissues are immersed in the Eb dye. Therefore, Eb staining can show cell death and assess the degree of cell membrane damage. As shown in Figure 2A, with the increase of concentrations of CeO_2_ NPs and Ce^3+^ ions, the fronds of duckweed were gradually stained with Eb, indicating that the cell activity decreased and oxidative stress increased. The quantitative results showed that the amount of Eb in the ionic group was about twice that in the NPs group at the highest concentration (Figure 2B), suggesting that the former caused more severe oxidative damage to duckweed. The staining of duckweed roots also showed concentration dependence, but there was no significant difference between the two treatments when the concentrations was lower than 0.5 mg/L (Figure 2C). At 10 mg/L Ce^3+^ ions, the root elongation was too short to be collected and measured.

### 3.5. SEM and TEM Observations

After exposure to 10 mg/L CeO_2_ NPs and Ce^3+^ ions, the morphological changes of duckweed fronds were observed via SEM. Compared to the control (Figure 3A), the frond area of the NP group was almost unchanged (Figure 3D), whereas that of the ionic group significantly decreased (Figure 3G). In addition, the stomata of the treated fronds were significantly deformed. The stomata of the frond in the control were intact and elliptical, with larger stomatal opening apertures. In contrast, the stomata in the NP group became narrower and the aperture became smaller (Figure 3F). The stomata in the ionic group were changed to be diamond-like, and the morphology of the guard cells was altered (Figure 3I). The aperture width and area of the duckweed stomata under the different treatments were calculated (at least 20 stomata), and the results are shown in Appendix A. The stomatal width and area in the NPs group were significantly lower than those of the control group, while the stomatal width and area in the ionic group were significantly higher than those of the control, which might be caused by the stomatal deformation under stress.

TEM images of duckweed root sections under different treatments are shown in Appendix A. The epidermis of the roots in the control group was smooth, the internal organelles were intact (Appendix A), and no deposits were found inside or outside the roots. In contrast, some particles were adsorbed on the root surface of duckweed in the CeO_2_ NP group, and some flocculent deposits were found in the internal intercellular spaces (Appendix A). Surprisingly, there were many needle-like deposits on the root surface of duckweed in the ionic group (Appendix A), and the morphology of the deposits was very similar to that of CePO_4_, which may be caused by the reaction of Ce^3+^ ions with phosphate in the culture medium. At the same time, high-electron-density substances were also found inside the cell (Appendix A), indicating that Ce^3+^ ions could be uptaken by duckweed roots and transformed into precipitates.

### 3.6. Elemental Mapping by μ-XRF

The distribution and localization of Ce, Ca, and Fe in the tissues are shown in Figure 4. After normalization, the minimum (blue) to maximum (red) values of fluorescence intensity were represented by color-coding diagrams. The high concentration of Ce in the NPs group was mainly accumulated at the edge of the fronds (Figure 4A), while that in the ionic group was mainly concentrated along the leaf veins (Figure 4B). Notably, strong Ce signals were observed at the root–frond junctions under both treatments, whereas no Ce signal was detected in the new frond of the ionic group (the upper-right corner of Figure 4B). In the roots, Ce signals in the NP-treated duckweed was mainly concentrated at the root tip (Figure 4C), while those in the ion-treated duckweed were detected mainly in the elongation region (Figure 4D). In addition, the intensity of Ce in the roots was much higher that in the fronds upon exposure to CeO_2_ NPs.

### 3.7. Contents and Speciation of Ce in Duckweed Tissues

Ce contents in duckweed exhibited dose-dependent manner under both treatments and increased with the increase of exposure concentration (Appendix A). Moreover, the Ce content in the ionic group was significantly higher than that of the NPs group at concentrations higher than 0.5 mg/L. The BAF values of Ce (defined as the ratio of Ce concentration in organisms to those in water) in duckweed decreased with increasing exposure concentrations in both groups (except 10 mg/L CeO_2_ NPs and 0.1 mg/L Ce^3+^ ions) (Appendix A). At each concentration, the BAF value of the ionic group was higher than that of the NP group. 

The speciation of Ce in the duckweed treated with 10 mg/L CeO_2_ NPs and Ce^3+^ ions was analyzed using XANES. In Figure 5, line a represents the characteristic peak of Ce(III), and lines b and c represent the characteristic peaks of Ce(IV). The spectrum of Ce in the sample of the NP group seemed to be a mixture of Ce(III) and Ce(IV) standard reference spectra. In contrast, the peak of Ce in the ionic group sample completely coincided with line a, suggesting that only Ce(III) existed. LCF results revealed that about 27.6% of Ce(IV) was reduced to Ce(III) in the NPs group, whereas 100% of Ce was present as Ce(III) in the ions group. 

## 4. Discussion

It has been reported that CeO_2_ NPs are non-toxic to several terrestrial plant species but specifically toxic to *Lactuca* plants [26], with the degree of inhibition being related to the species and the type of culture medium [27]. However, there are few reports about the effects of CeO_2_ NPs on the growth of aquatic plants. In the present study, we compared the toxicity of CeO_2_ NPs and Ce^3+^ ions to the aquatic plant duckweed at concentrations ranging from 0 to 10 mg/L. The significant inhibitory effect of CeO_2_ NPs on the growth of duckweed at such a low concentration (0.1 mg/L) was surprising, indicating that duckweed is very sensitive to non-environmental stresses (such as exogenous NPs) and can be used as an indicator of environmental pollution. The toxicity of Ce^3+^ ions to duckweed was greater than that of CeO_2_ NPs at the same concentration. This was similar to the results of a previous study, in which the authors found that ionic cerium (Ce^3+^ ions) at 10 mg/L had a negative effect on the growth of radish, while bulk and nanoparticulate CeO_2_ at the same concentration had no effect on radish growth and increased plant biomass, respectively [28].

Many physiological and biochemical activities during plant growth, such as chloroplast synthesis and photosynthesis are associated with the uptake and utilization of essential elements [29,30]. Deficiencies of certain mineral elements may affect plant respiration and photosynthesis, resulting in growth inhibition [31]. The mineral element contents of duckweed were disturbed under both CeO_2_ NPs and Ce^3+^ ions treatments (Table 1). The effective ion radius of REE ions are close to that of Ca^2+^, they may compete with Ca for organic ligands or replace it in the cell walls [32]. The decrease in Ca content may be due to the replacement of Ca^2+^ by Ce^3+^ ions in the cell wall. Moreover, the decrease in Ca content may affect membrane stability, leading to a decrease in chlorophyll content [33]. Therefore, the reduction of photosynthetic pigments induced by CeO_2_ NPs and Ce^3+^ ions in this study might be related to the decrease of Ca content. Fe is an activator of coenzymes related to the synthesis of chlorophyll, and the decrease in Fe will reduce the content of chlorophyll [34,35], which might also be one of the reasons for the reduction in photosynthetic pigments in the treated duckweed. It has been reported that rare earth elements (REEs) can regulate plant growth by influencing the mineral elements in the plant tissues [36], and the uptake of Ca, Na, Zn and Mn by corn and mung bean decreased with increases in La^3+^ and Ce^3+^ ion concentrations (0–5 μM) in the exposed medium [37]. Similarly, reductions in the content of related elements and corresponding growth inhibition were observed in duckweed treated with CeO_2_ NPs and Ce^3+^ ions.

The state of stomata can reflect the growth of plant leaves, and photosynthesis is sensitive to the disturbance of gas exchange through stomata. The decrease of stomatal aperture may result in the reduction in net photosynthesis and transpiration rate [38]. In this study, duckweed stomata were deformed when exposed to both CeO_2_ NPs and Ce^3+^ ions, indicating that the morphology and function of stomata were severely damaged. Similar stomatal closure was also observed in the fronds of purple-backed duckweed exposed to Ag NPs [16] and *Eichhornia crassipes* exposed to CuO NPs [39]. The authors attributed this alteration to the overproduction of ROS. In this study, the results of Eb staining indicated that both CeO_2_ NPs and Ce^3+^ ions produced excessive ROS in duckweed at high concentrations, which led to cell death and affected plant physiological characteristics, such as root elongation, frond area, and morphology. 

The different in situ distributions of Ce in duckweed tissues under the two treatments implied that the uptake and transport of the two materials might be different. The distribution of Ce in duckweed fronds was similar to that previously reported in cucumber—namely, Ce mainly concentrated at the edge of CeO_2_ NPs-treated leaves, whereas Ce in the Ce^3+^-ion-treated leaf mainly concentrated along the veins [20]. Although both roots and fronds of duckweed were in contact with CeO_2_ NPs (or Ce^3+^ ions) upon exposure, the distribution of Ce in duckweed fronds was similar to that in cucumber leaves under the same treatment, indicating that roots play an important role in the accumulation of Ce in duckweed. In the roots, the Ce signal in the NPs group was concentrated at the root tip and elongation region, while that in the ionic group was mainly detected in the elongation region. Plant roots can secrete mucilage containing organic acids, which can promote the adsorption and internalization of NPs. One of the possible ways by which NPs enter plants is from the root meristem entering into the vascular bundle through the apoplast; they are then transported to shoots through transpiration [20]. The accumulation of Ce in the root tips of duckweed exposed to CeO_2_ NPs in this study also suggested this possibility. In addition, high-Ce signals were observed at the root and frond junctions under both treatments, while no Ce signals were detected in the new fronds. This suggested that Ce might be taken up by the roots and then transported upward to the fronds by transpiration with the water flow rather than being absorbed directly through the fronds. This result was consistent with the accumulation of Cu in duckweed exposed to CuO NPs being mainly attributed to the roots [19].

Due to the highly reactive and dynamic properties of NPs, they may undergo transformation upon interaction with plants [40]. Since the “acquired” and “original” characteristics of NPs coexist in the real environment, relevant factors must be considered when assessing their environmental fate and biological effects [41]. CeO_2_ NPs were originally thought to be stable and insoluble compounds in the environment. However, later studies proved that dissolution of CeO_2_ NPs did occur under certain circumstances [42]. For example, when CeO_2_ NPs interact with terrestrial plants, they can be transformed and release Ce(III) ions with the assistant of root exudates [43,44]. Root exudates contain a variety of low molecular compounds, such as organic acids, amino acids, carbohydrates, polysaccharides, and phenolic compounds, which were critical for the dissolution and uptake of CeO_2_ NPs by plants. Similar to terrestrial plants, aquatic plants have a special root-affected area, where root exudates interact with NPs [45]. Root exudates of aquatic plants mainly consist of carbon-containing compounds, such as organic acids, amino acids, carbohydrates, polysaccharides, proteins, tannins, phenolic compounds, etc. [46]. As we demonstrated, CeO_2_ NPs could be reduced to Ce(III) in aquatic plant duckweed with the assistant of organic acids and reducing substances (such as phenols). The degree of transformation of CeO_2_ NPs (the amount of ions released) upon interacting with plants varied from species to species, and the phytotoxicity of CeO_2_ NPs was mainly plant-species-dependent. For instance, the phytotoxicity of CeO_2_ NPs to *Lactuca* plants was due to the high sensitivity of this species to Ce^3+^ ions produced through dissolution after the interaction between them [26,27]. According to the proportion of Ce(III) and the total Ce content in duckweed treated with CeO_2_ NPs (10 mg/L), the absolute content of Ce(III) was calculated to be 948.0 mg/kg, which was comparable to that in duckweed treated with Ce^3+^ ions of 2 mg/L. By retrieving the apparent toxicity parameters (Figure 1), it was found that the inhibition degree of 10 mg/L CeO_2_ NPs on duckweed was exactly between that of 0.5 and 2 mg/L Ce^3+^ ions. Therefore, we speculated that the phytotoxicity of CeO_2_ NPs to duckweed was due to the transformation and the high sensitivity of duckweed to the released Ce^3+^ ions.

## 5. Conclusions

In summary, the effects of CeO_2_ NPs and Ce^3+^ ions on the aquatic higher plant duckweed were studied. At concentration higher than 0.1 mg/L, they could reduce the biomass and photosynthetic pigment contents of duckweed, causing significant cell death. Moreover, the toxicity of Ce^3+^ ions to duckweed was greater than that of CeO_2_ NPs of the same concentration. SEM observation showed that the frond stomata were deformed, and the parameters of stomatal morphology were significantly changed under the two treatments. The in situ distribution of Ce in duckweed roots suggested that roots played a dominant role in the uptake of Ce. XANES and LCF results confirmed the transformation of Ce from Ce(IV) to Ce(III) in CeO_2_-NP-treated duckweed and that the toxicity of CeO_2_ NPs to duckweed originated mainly from the sensitively of this species to the released ions. The findings of this work were helpful for assessing the ecotoxicity of CeO_2_ NPs and for better understanding the interaction between metal-based NPs and aquatic plants.

## Figures and Tables

**Figure 1 nanomaterials-13-02523-f001:**
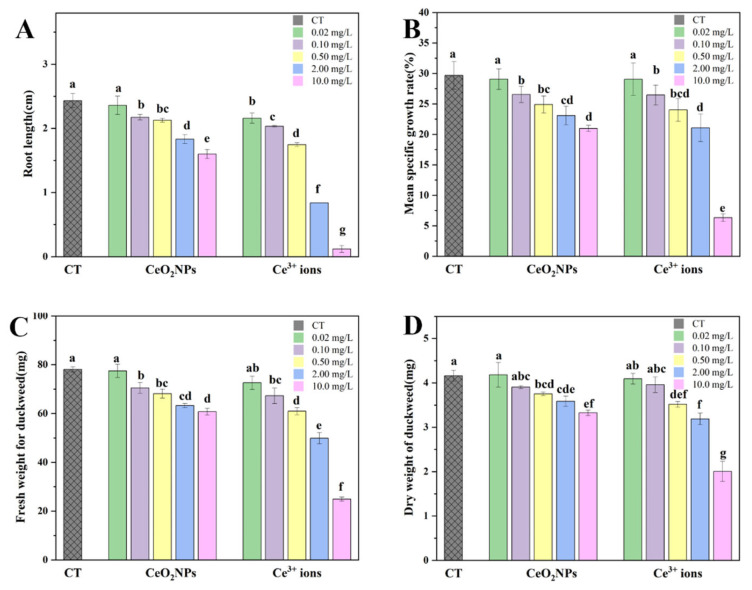
Root length (**A**), mean specific growth rate (**B**), fresh weight (**C**), and dry weight (**D**) of the duckweed treated with CeO_2_ NPs and Ce^3+^ ions. Different letters indicate significant differences (*p* < 0.05).

**Figure 2 nanomaterials-13-02523-f002:**
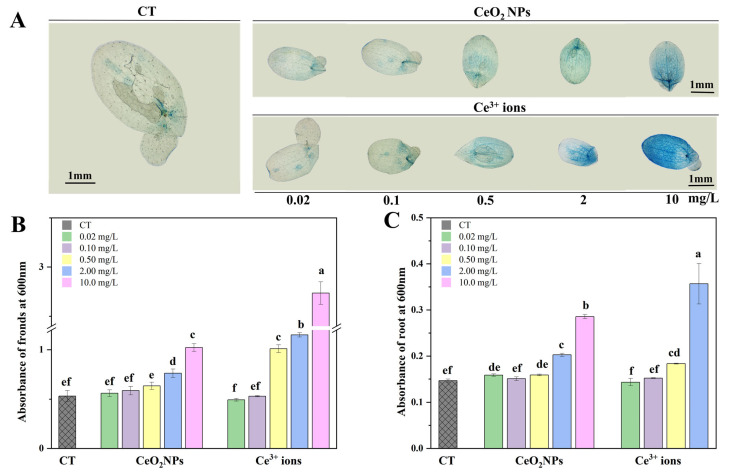
Visualization of cell death in fronds stained with Eb (**A**). The absorbance values of Eb in fronds (**B**) and roots (**C**) of duckweed under different treatments. Different letters indicate significant differences (*p* < 0.05).

**Figure 3 nanomaterials-13-02523-f003:**
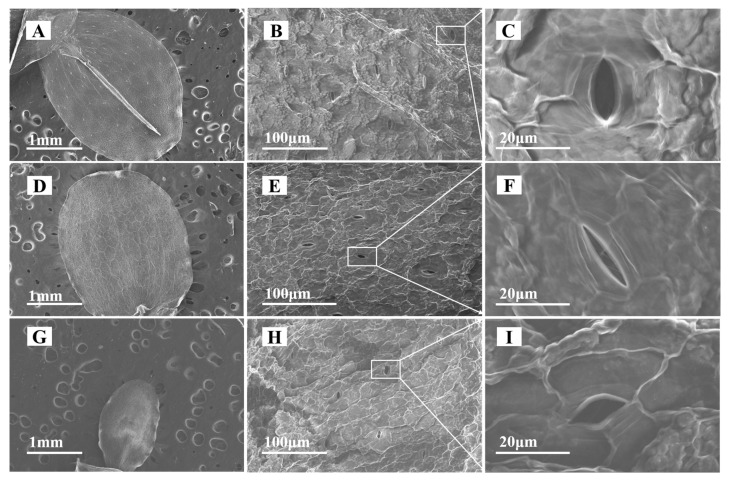
SEM observations of fronds and stomatal morphology of duckweed unexposed (**A**–**C**), exposed to 10 mg/L CeO_2_ NPs (**D**–**F**) and exposed to10 mg/L to Ce^3+^ ions (**G**–**I**). Panels (**C**,**F**,**I**) are enlarged from panels (**B**,**E**,**H**), respectively.

**Figure 4 nanomaterials-13-02523-f004:**
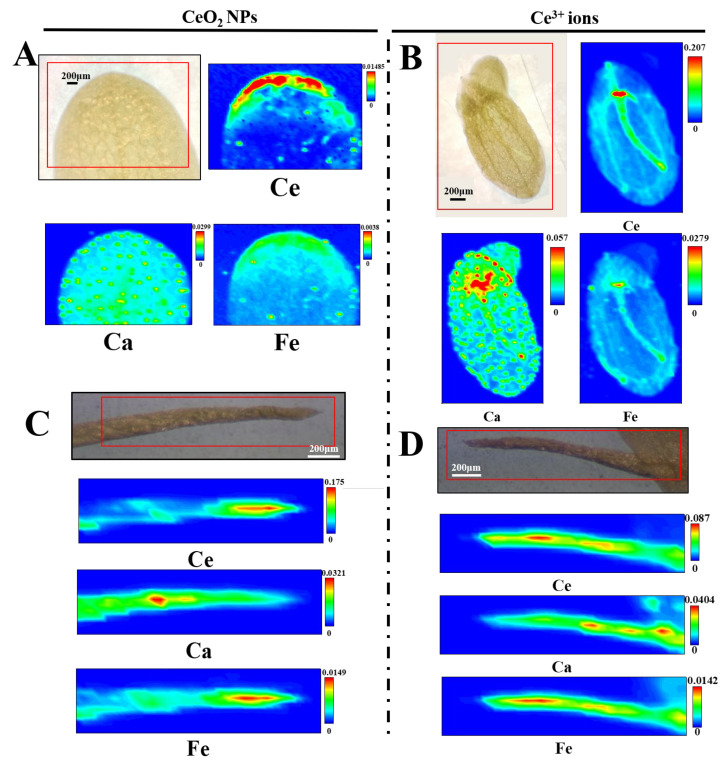
*μ*-XRF images of Ce, Ca, and Fe in the fronds (**A**,**B**) and roots (**C**,**D**) of duckweed exposed to 10 mg/L CeO_2_ NPs (**A**,**C**) and Ce^3+^ ions (**B**,**D**). The red color in each image corresponds to the maximum concentration of the element.

**Figure 5 nanomaterials-13-02523-f005:**
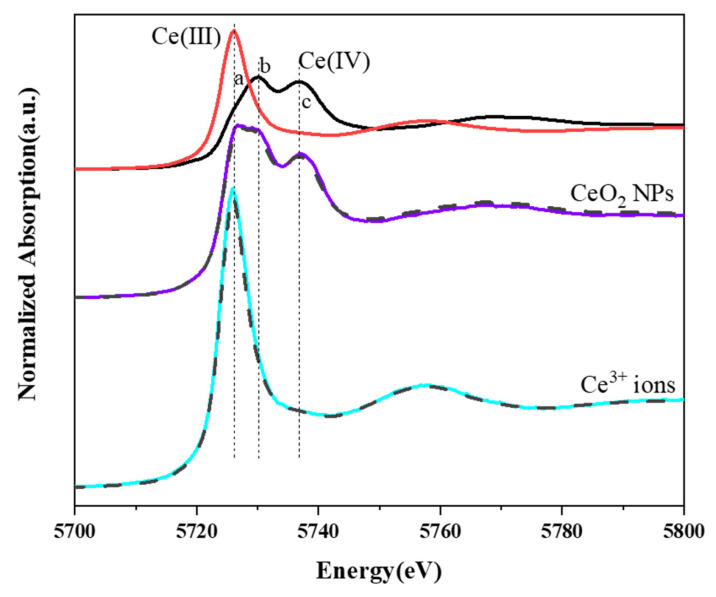
XANES spectra of Ce L_III_-edge in duckweed treated with 10 mg/L CeO_2_ NPs and Ce^3+^ ions.

**Table 1 nanomaterials-13-02523-t001:** Mineral element concentrations in duckweed after exposure to CeO_2_ NPs and Ce^3+^ ion ^1^.

Treatments (mg/L)	Macroelements (μg/g·Dw)		Microelements (μg/g·Dw)		
	Ca	Mg	Fe	Mn	Zn
0 (CT)	6209.1 ± 72.7 a	2249.91 ± 70.6 a	168.3 ± 14.3 a	98.8 ± 3.9 a	30.4 ± 5.3 a
CeO_2_ NPs					
0.02	5464.7 ± 70.2 b	2312.4 ± 143.3 a	135.48 ± 5.5 b	92.4± 0.7 b	24.1 ± 9.6 ab
0.1	5191.4 ± 114.7 bc	2227.3 ± 78.2 a	143.0 ± 35.4 ab	85.5 ± 1.5 c	19.8 ± 5.9 abc
0.5	5234.3 ± 295.2 bc	2343.4 ± 93.1 a	96.1 ± 1.444 c	83.4 ± 1.7 c	11.4 ± 0.7 bc
2	4948.5 ± 89.6 c	2401.45 ± 17.1 a	83.3 ± 6.8 c	84.6 ± 1.7 c	15.4 ± 4.3 bc
10	4970.2 ± 172.0 c	2265.7 ± 47.8 a	83.8 ± 1.4 c	86.2 ± 2.0 c	7.3 ± 0.7 c
Ce^3+^ ions					
0.02	5606.9 ± 146.3 b	2188.8 ± 93.7 a	125.3 ± 16.6 b	87.8 ± 1.6 a	12.3 ± 5.0 b
0.1	5128.4 ± 211.8 c	2254.2 ± 196.3 a	99.7 ± 0.5 b	88.4 ± 3.8 a	22.9 ± 3.0 ab
0.5	5262.0 ± 221.0 bc	2273.8 ± 69.5 a	108.6 ± 2.3 b	82.6 ± 2.0 a	13.2 ± 4.3 b
2	5037.6 ± 132.1 c	2652.4 ± 542.3 a	124.6 ± 29.4 b	100.6 ± 25.4 a	21.3 ± 4.8 ab
10	3502.1 ± 153.6 d	929.2 ± 24.4 b	133.9 ± 11.7 b	51.5 ± 1.3 b	17.2 ± 4.2 b

^1^ Data are average of three replicates ± SD. Different letters among columns indicate significant difference at *p* < 0.05.

## Data Availability

Not applicable.

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
