# Peer review of "Interaction of Cerium Oxide Nanoparticles and Ionic Cerium with Duckweed (Lemna minor L.): Uptake, Distribution, and Phytotoxicity"

_nanomaterials, 2023, doi:10.3390/nano13182523_

Round 1
Reviewer 1 Report
This research was done qualitatively. May be accepted for publication in this form
Author Response
Response: Thanks for your comments and suggestion.
Reviewer 2 Report
I have read with interest the research paper of Liu et al. on some aspects of the eco-toxicity of CeO2 nanoparticles. The manuscript is well-written, easy to follow, lack of edition mistakes and both the title and the abstract clearly present the question posed by the authors. The experimental methodology is also well described and easy to reproduce in other laboratories. Specifically, the description of the results and the figures are clear and simple enough to get easily the information the reader may be interested; and I enjoyed reading it. Moreover, for the importance of CeO2NPs in the current research in biomedicine and catalysis, the data here presented is a good contribution in order to understand eco-safety aspect of the use of these materials. A few considerations:
1. Given that the nanomaterials are known to change when exposed to media that is different from the synthesis media, I would suggest to add some TEM images of the NPs before and after exposure. Also to provide the DLS graph before and after (not only the values)
2. Authors observe certain agglomeration of the CeO2NPs in the modified Steinberg nutrition medium. Can agglomeration contribute to observed toxicity at high concentration? Maybe it can be discussed
3. Are the concentration of Ce3+ and CeO2NPs environmentally relevant or realistic?
Author Response
Response: Thanks for your positive comments and thoughtful suggestions for improving this manuscript. Detailed responses are provided below.
- Given that the nanomaterials are known to change when exposed to media that is different from the synthesis media, I would suggest to add some TEM images of the NPs before and after exposure. Also to provide the DLS graph before and after (not only the values).
Response: Thanks for your comments. TEM images (Figure S1 and S8) and DLS graphs (Figure S2 and S3) of the NPs before and after exposure have been added in the revised version.
- Authors observe certain agglomeration of the CeO2 NPs in the modified Steinberg nutrition medium. Can agglomeration contribute to observed toxicity at high concentration? Maybe it can be discussed.
Response: Thanks for your comments. The degree of dispersion or aggregation of nanoparticles may affect their bioavailability, thereby affecting their toxicity. Burke et al. found that Fe3O4 NPs with good dispersibility are more easily uptake by plants compared to TiO2 NPs that form larger aggregates (Iron oxide and titanium dioxide nanoparticle effects on plant performance and root associated microbes. Int. J. Mol. Sci. 2015, 16, 23630–23650.). Konate et al. found that low concentrations (50 mg/L) of Fe3O4 NP were more toxic to cucumbers than high concentrations (500 and 2000 mg/L), possibly due to the better dispersity of the former (Comparative effects of nano and bulk-Fe3O4 on the growth of cucumber (Cucumis sativus). Ecotoxicology and Environmental Safety, 2018, 165, 547-554.). In this study, we found certain agglomeration of the CeO2 NPs (10 mg/L) in the modified Steinberg’s medium, which might affect their bioavailability to duckweed and underestimated their toxicity than the well dispersed one.
- Are the concentration of Ce3+ and CeO2 NPs environmentally relevant or realistic?
Response: Thanks for your comments. Because there is no research on the biological effects of CeO2 NPs on duckweed, we conducted preliminary experiments first. Based on the preliminary experimental results, a series of concentrations of CeO2 NPs were set for exposure to duckweed, and corresponding Ce3+ ions treatments were also set for comparative study. The minimum exposure concentration was 20 μg/L, which was comparable to the reported environmental concentration, i.e., the predicted environmental concentrations (PEC) of CeO2 NPs in freshwater surfaces and sediments were 0.51-25.59 ng/L and 0.34-253.49 μg/kg, respectively. We found that CeO2 NPs significantly inhibited the root elongation of duckweed at concentrations as low as 100 μg/L, while the inhibition threshold of Ce3+ ions was 20 μg/L.
Reviewer 3 Report
Introduction:
Add reference” The global production of CeO2 NPs was estimated about 10,000 tons per year.”
Revise”, use and disposal of NPs, they (including CeO2 NPs) ”
Add reference” . The predicted environmental concentrations (PEC) of CeO2 NPs in fresh water and sediments were 0.51-25.59 ng/L and 0.34-253.49 μg/kg in 2017, respectively.”
Add few references to “Currently, there are few studies on the effects of CeO2 NPs on aquatic plants”
Write the full name for the “OECD”
Authors can benefit from citing these relevant article which investigated nanoparticles including CeO2
Impact of nanoparticle surface charge and phosphate on the uptake of coexisting cerium oxide nanoparticles and cadmium by soybean (Glycine max. (L.) merr.)
International Journal of Phytoremediation 22 (3), 305-312
Aerially applied zinc oxide nanoparticle affects reproductive components and seed quality in fully grown bean plants (Phaseolus vulgaris L.)
Frontiers in Plant Science, 3097
Zinc oxide (ZnO) nanoparticles elevated iron and copper contents and mitigated the bioavailability of lead and cadmium in different leafy greens
Ecotoxicology and Environmental Safety 191, 110177
Silver nanoparticle detection and accumulation in tomato (Lycopersicon esculentum)
Journal of Nanoparticle Research 22, 1-16
The English is good enough. However, the introduction needs some minor editions.
Author Response
Introduction:
Add reference” The global production of CeO2 NPs was estimated about 10,000 tons per year.”
Response: Thanks for your comments. It has been added.
Revise”, use and disposal of NPs, they (including CeO2 NPs) ”
Response: Thank you. This sentence has been rewritten.
Add reference” . The predicted environmental concentrations (PEC) of CeO2 NPs in fresh water and sediments were 0.51-25.59 ng/L and 0.34-253.49 μg/kg in 2017, respectively.”
Response: It has been added.
Add few references to “Currently, there are few studies on the effects of CeO2 NPs on aquatic plants”
Response: It has been added.
Write the full name for the “OECD”
Response: The full name for the “OECD” has been added.
Authors can benefit from citing these relevant article which investigated nanoparticles including CeO2
Impact of nanoparticle surface charge and phosphate on the uptake of coexisting cerium oxide nanoparticles and cadmium by soybean (Glycine max. (L.) merr.)
International Journal of Phytoremediation 22 (3), 305-312
Aerially applied zinc oxide nanoparticle affects reproductive components and seed quality in fully grown bean plants (Phaseolus vulgaris L.)
Frontiers in Plant Science, 3097
Zinc oxide (ZnO) nanoparticles elevated iron and copper contents and mitigated the bioavailability of lead and cadmium in different leafy greens
Ecotoxicology and Environmental Safety 191, 110177
Silver nanoparticle detection and accumulation in tomato (Lycopersicon esculentum)
Response: Thanks for your suggestions. We have read these relevant articles and cited them where appropriate in the revised version.
Comments on the Quality of English Language
The English is good enough. However, the introduction needs some minor editions.
Response: Thanks for your positive comments and thoughtful suggestions for improving this manuscript.